

# Ancient clays support contemporary biogeochemical activity in the Critical Zone

Vanessa M. Alfonso[1], Peter M. Groffman[1,2], Zhongqi Cheng[1], David E. Seidemann[1]

[1] Department of Earth and Environmental Sciences, Brooklyn College of the City University of New York, 2900 Bedford Avenue, Brooklyn, NY 11210, USA

[2]Advanced Science Research Center at the Graduate Center, City University of New York, 85 St. Nicholas Terrace, New York, NY 10031 USA

*Correspondence to: Vanessa.Alfonso23@bcmail.cuny.edu*



**Abstract**

Late Cretaceous clays exposed at sites located on the north shore of Long Island, New York, USA were sampled to explore questions about how contemporary factors and processes interact with ancient geological materials. Chemically and biologically catalyzed weathering processes have produced multi-colored clays belonging to the kaolin group with inclusions of hematite, limonite, and pyrite nodules. We sampled exposed clays at three sites to address three questions: 1) Do these exposed clays support significant amounts of microbial biomass and activity, i.e., are they alive? 2) Do these clays support significant amounts of nitrogen (N) cycle activity? 3) Are these clays a potential source of N pollution in the contemporary landscape? Samples were analyzed for total carbon (C) and N content, microbial biomass C and N content, microbial respiration, organic matter (OM) content, potential net N mineralization and nitrification, soil nitrate ($NO_3^-$) and ammonium ($NH_4^+$) content, and denitrification potential. Results strongly support the idea that ancient geologic materials play a role in contemporary N and C cycling in the Critical Zone. Respiration was detectable in all samples and was strongly correlated to OM, indicating a living microbial community on the clays. There was evidence of an active N cycle. Higher levels of denitrification potential compared to both potential net nitrification and potential net N mineralization indicate that these clays act more as a sink rather than as a source of N pollution in the landscape.

## 1 Introduction

The Critical Zone is Earth's constantly evolving boundary layer where rock, soil, water, air, and living organisms interact (Schroeder, 2018). It is comprised of the solid phase matter in which water circulates and is stored, expanding from the top of the vegetation canopy down into the water bearing bedrock. Critical Zone processes are key drivers of chemical transfers between biota and geological materials (Brantley et al., 2006). A major question in Critical Zone science is how contemporary factors and processes interact with ancient geological materials. These interactions are influenced by the geochemical composition and pore space of these materials, which affect microbial activity and therefore rates of biogeochemical processes in the carbon (C) and nitrogen (N) cycles (Li et al., 2023). These interactions are particularly obvious and important in the lower boundary of the Critical Zone, and in places where ancient geologic materials become exposed to contemporary environments. Major questions center on the ability of ancient materials to support biogeochemical processes related to the cycling of C and N that underlie plant and microbial activity, which underlies environmental and ecosystem "services" of interest to society.

The recent discovery of significant amounts of N in sedimentary deposits increased interest in the role of geological materials in contemporary N cycling (Morford et al., 2011). Up to 17% of the currently cycling N in some ecosystems may originate from rock materials deep in the Critical Zone (Houlton et al., 2018). Such observations motivate the analysis of surface-exposed ancient rocks in this study.



We measured microbial biomass and activity in clays exposed in outcrops with a focus on C and N cycle processes to address three questions: 1) Do these exposed clays support significant amounts of microbial biomass and activity, i.e, are they alive? To address this question we measured microbial biomass C, an index of the living microbial biomass in soil, and microbial respiration, a direct measure of microbial activity (Paul, 2014). Total C and organic matter (OM) content were measured as energy sources for microbial biomass and activity. 2) Do these clays support

significant amounts of N cycle activity? N cycle activity was assessed with measurements of microbial biomass N content, potential net N mineralization, and total N content. Microbial biomass N provides an index of the net flux of N through microbial pools. Mineralization results from microbial degradation of N compounds resulting in the production of inorganic, plant-available forms of N. Total N content was measured to quantify the total amount of N potentially available for active cycling. 3) Are these clays a potential source of N pollution in the contemporary

landscape? The potential for the clays' microbial activity to be a source or "sink" for N pollution was evaluated by measuring potential net nitrification, denitrification potential, and pools of ammonium ($NH_4^+$) and nitrate ($NO_3^-$). Both $NH_4^+$ and $NO_3^-$ are highly soluble and the negative charge of $NO_3^-$ makes it highly mobile, thus driving hydrologic losses of N. In excessive amounts, these nutrients lead to ecological stresses such as eutrophication. $NH_4^+$ is converted to $NO_3^-$ by nitrifying bacteria, and $NO_3^-$ is converted to $N_2$ by denitrifying bacteria leading to

gaseous losses of N back into the atmosphere (Seitzinger et al., 2006).

## 2 Background

    Silicates are the most abundant mineral group on Earth. Clay minerals are hydrous alumino-silicates, more

specifically hydrous phyllosilicates, and are some of the most stable products of chemical weathering at surface conditions. Clay minerals make up approximately 40% of the minerals in sedimentary rocks, and about 16% of the solid part of the Critical Zone. They are the constituents of soils, mudstones, claystones, and shales (Schroeder, 2018). Clay minerals produce a specialized microhabitat and their ability to store and release nutrients make them ecologically important (Kleber et al., 2021). Clay minerals have high surface adhesion capabilities and sorption and

desorption of OM in soils varies with mineral assemblage. Sorption of organic carbon (OC) onto phyllosilicates and hydrous iron (Fe) oxides affects accumulation and stabilization of OC in soils (Saidy et al., 2013). Ecosystem function is greatly influenced by mineral abundances. The 1:1 layers of hydrated kaolinite clays and 2:1 layers of mixed layer clays serve as nutrient exchange sites between biomass and subsurface weathering horizons. Therefore, the layered structures of clays are key facilitators of seasonal cycling of nutrients including $NH_4^+$ among others

(Eby, 2016; Halama and Bebout, 2021). Chemical weathering of silicate minerals is a significant mechanism for the availability, uptake, storage, and transport of key nutrients in ecosystems. Variations in mineral weathering and nutrient availability occur due to microorganism and mineral speciation, while intensity of mineral weathering is influenced by a mineral's potential to provide nutrients (P. C. Bennett et al., 2001). Nutrients are extracted from rocks by weathering processes since rocks are the primary source of nutrients except C and N. The pathways of C

and N from the atmosphere to incorporation into the lithosphere progresses from fixation into OM to storage in low



temperature silicate phases such as clay minerals (Busigny and Bebout, 2013; Halama and Bebout, 2021). This is facilitated by the aqueous solutions and living microbes characteristic of the Critical Zone.

Late Cretaceous clays on the north shore of Long Island, New York, USA offer an opportunity to study the effects
of contemporary factors and processes on ancient Critical Zone materials. Long Island is composed of Pleistocene sediments deposited on top of Late Cretaceous formations (Sirkin, 1991) and includes aquifers, confining units, and clay deposits such as the Raritan, Magothy, Gardiners, and Wantagh Clay Formations. Coastal exposures of the Raritan and Magothy formations present opportunities for biogeochemical analysis of these materials. At these locations clays are exposed in outcrops, offering the chance to investigate their contemporary biogeochemical
activity at surface conditions (Fig. 1). Long Island's clay strata have been mainly accessed through core drilling for hydrological studies, and in most instances the clays are a secondary detail rather than the main subject of study. The same is true for palynology studies in which the clay is the assemblage containing the ancient pollens being studied. In contrast, this study focuses on biogeochemical activity and how the clay strata affect and interact with surrounding environments in the Critical Zone. The kaolinitic materials at these sites and their abundance in oxides
contribute to long lasting micro-environments (Six et al., 2000).

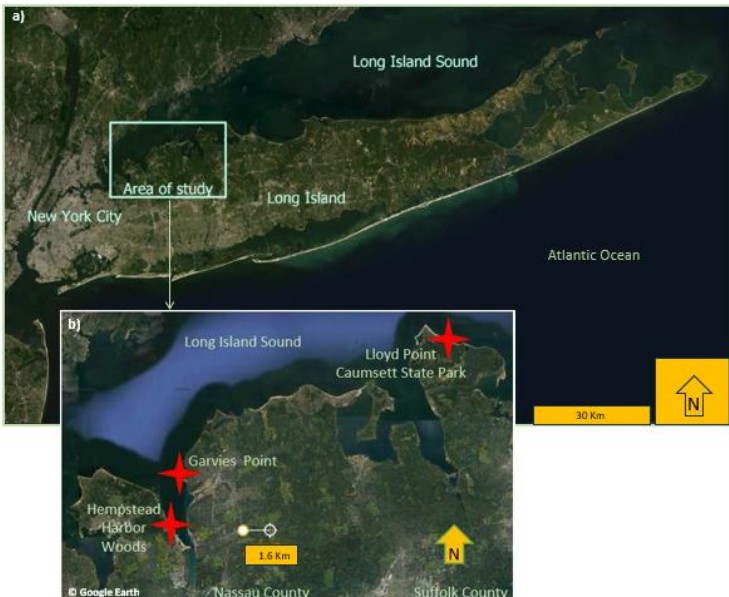

**Figure 1: Map of study sites on Long Island, New York. Map data: a) Esri, b) Google Earth.**

All three study sites are located on the north shore of Long Island, NY. Two of the sites are located along the shoreline - Garvies Point Preserve (GP) on Hempstead Bay and Caumsett State Historic Park Preserve (CSP) along
Long Island Sound; Hempstead Harbor Woods (HHW) is located inland from the western side of Hempstead Bay. Long Island contains numerous clay beds such as Gardiner's Clay, Raritan Confining Unit, Wantagh Clay, and Smithtown Clay, as well as clay lenses in the Magothy Formation (Mills and Wells, 1974). The study area was



shaped by the Wisconsin glaciation approximately 75,000 to 11,000 years ago. During this period, Cretaceous strata were sheared off, transported, and re-deposited. Glacially induced thrusting of the strata facilitated bulging of the

clay (Mills and Wells, 1974), thus leading to eventual exposure. Evidence of this can be observed in the outcrop sampled for this study at GP, where the clay layers are currently oriented vertically rather than in a near horizontal depositional position. Although there is consensus that the Late Cretaceous deposits of the western north shore of Long Island were formed in a shallow delta or estuary (Fuller, 1914; Swarzenski, 1963) and that they are composed of sand, silt, gravel, and clay, further augmented by eolian action, it is not completely clear to which formation the

exposed North Shore clays belong. It is generally accepted that the Late Cretaceous formations exposed on the North Shore tentatively belong to the Magothy formation but may include some younger formations (Isbister, 1966). It is presumed that Pleistocene deep permafrost formation must have preceded glacial thrusting (Mills and Wells, 1974), which provided the sedimentary cohesion needed to produce the tilted clay strata and layered shale observed today. The three sites of this study also differ in their position on the Manhasset Plateau. The HHW and GP

localities are situated on the Upper Manhasset Plateau while CSP is situated on the Lower Manhasset Plateau. The Lower Manhasset Plateau is thought to have sustained a more prolonged grinding by the ice sheet (Fuller, 1914). The most abundant clay species identified from core drills in northwestern Long Island is kaolinite of light gray, brown-yellow, and tan color, along with sparse chlorite, vermiculite, and montmorillonite, with kaolinite ratio to other clays decreasing with depth (Liebling, 1973).

Fe content, among other factors, has contributed to variations in coloring and produced distinctive clay strata, allowing for color based grouping of the samples. The color variations are the result of differences in constituents and geochemical processes that evolved the clays into their modern state. The Fe oxides in hematite are likely sources of the red and brownish coloring and the Fe oxides in goethite are likely sources of the yellow coloring (Davey et al., 1975). This is further supported by pXRF scan results (sections 4 and 5) which show red, yellow, and

brown samples to have some of the highest Fe content. Mn also contributes to brown coloring (Jakobsson et al., 2000), and this is also further supported by pXRF scans that show brown samples to have the highest Mn content. Recent conditions have added sand and water to some clays to yield different textures, providing another observational classification criterion.

**3 Methods**


**3.1 Study sites**

Garvies Point Preserve (GP) is located in Glen Cove, NY, on the eastern shoreline of Hempstead Bay at 40°51'35" N 73°39'07" W. The exposed clays were accessed via trails leading to the beach and samples were collected along

approximately 450 m of shoreline, with special focus on the main outcrop which features four types of clays exposed at this site. The most prominent outcrop at GP is approximately 4 m high and yielded samples from five differently colored clays (light gray, dark charcoal gray, white, yellow, and dark red/purplish). The adjacent outcrop is approximately 2 m high containing light and dark gray clay.



Hempstead Harbor Woods (HHW) is located in North Hempstead, NY, at 40°50'11" N 73°39'57" W, on the inland
       western side of Hempstead Bay, approximately 300 m from the shoreline. Samples were collected from 0 – 1 m
       from the ground level throughout the wooded area. Although this location does not have direct exposures to the bay,
       some of the sampled clay materials have indicators of being part of the same formation as those exposed across the
       bay at GP. Specifically, the red and light gray packed clays collected at HHW share the color and texture attributes
of those at GP.

       At Caumsett State Historic Park Preserve (CSP) samples were collected from exposures along approximately 1000
       m of shoreline. A prominent exposure at CSP located at 40°56'21" N 73°28'13" W has an elevation of
       approximately 40 m. Samples were collected starting from the bottom of the cliff, just above the beach floor, and as
high as 6 m above the beach floor. Additional samples were collected from exposures at 40°56'08" N 73°28'14" W
       approximately 4 m above the beach floor, and at 40°56'11" N 73°28'43" W approximately 6 m above the beach
       floor.

       **3.2 Sampling**

At each site samples were collected from locations that indicated the highest likelihood of yielding multiple clay
       types. Large outcrops containing multiple variations of clay at GP and CSP were sampled more extensively. The
       prominent outcrops at these locations feature variety in both color and texture. HHW featured scattered locations,
       each featuring clay of a single texture and color. One sample from HHW (sample 3N-HHW) and one sample from
       CSP (sample 1N-CSP) were collected from naturally exposed soil horizons (B horizon) underneath topsoil (O
horizon).

       Samples were collected from the surface of the exposures. Locations where clay had at least a few inches of depth
       were chosen. The uppermost layer (~1 cm) was scraped off to ensure only clay was collected and to remove any
       field debris such as loose soil, rocks, sand, and loose plant material. Samples were collected from approximately 5-
       10 cm of depth. The samples were collected using a trowel, placed in labeled Ziplock bags, and packed loosely with
some air remaining in the bag. Rocks, roots, and leaves were removed by hand right after collection. The samples
       were then refrigerated until laboratory processing. Samples were classified into groups of color (brown, dark grey,
       light gray, red, white, yellow) and texture (packed clay, sandy clay, watery clay) by visual observation. The packed
       clay was homogeneous, contained some moisture, and had a putty like texture that easily formed into a ribbon
       several inches long. The sandy clay was drier and had fine sand mixed in. The watery clay was collected from a
small basin of waterlogged clay that was homogeneous, had very fine particles, and a thick viscous texture.



### 3.3 Laboratory analysis


Samples were analyzed for moisture content by oven drying for 24 hours at 105°C. Moisture content was used to calculate values for all variables on a per g of dry soil basis.

OM content was determined by Loss on Ignition (LOI). Oven dried samples were "ignited" at 450°C in a muffle furnace and % OM was calculated from weight loss after 8 hours of heating.

For elemental analysis by pXRF, samples were oven dried for 24 hours at 80°C, ground with a mortar and pestle, sieved through a #230 (63μm) sieve, and scanned with a portable Olympus DC-4000 XRF scanner.

Total C, total N, and the C/N ratio were measured using flash combustion / oxidation. Oven dried and ground samples were pressed into 1g pellets. The pellets were analyzed in an Elementar varioMax Cube elemental analyzer.

For analysis of exchangeable $NO_3^-$ and $NH_4^+$, samples were blended with 2M KCl on an orbital shaker at 125 rpm for one hour, followed by filtration (Whatman #42 filter) into scintillation vials that were immediately refrigerated until analysis. The samples were pipetted into microplates and analyzed on a SpectraMax M2 Multi-Mode Microplate Reader from Molecular Devices using wavelengths of 450 nm for $NO_3^-$, and 650 nm for $NH_4^+$ (Doane and Horwáth, 2003; Sims et al., 1995). Total inorganic N (TIN) was calculated as the sum of $NH_4^+$ and $NO_3^-$.

The chloroform fumigation and incubation method (CFIM) (Jenkinson and Powlson, 1976) was used to determine the C and N content of microbial biomass. Samples (10 g) were fumigated with chloroform for 24 hours, inoculated with 0.2 g of unfumigated clay and incubated for 10 days in 946 ml "mason" jars with lids fitted with septa. At the end of the incubation, gas samples were taken by syringe and analyzed for carbon dioxide ($CO_2$) with a Shimadzu GC-2014 gas chromatograph. After gas sampling, fumigated soils were extracted using KCl as described above

after 10 days and the $NO_3^-$ and $NH_4^+$ produced over the 10 day incubation was taken as an estimate of microbial biomass N. A proportionality constant of 0.41 was used to calculate microbial biomass C from the $CO_2$ produced over the 10 day incubation.

Unfumigated samples were also incubated for 10 days and provided estimates of microbial respiration and potential net N mineralization and nitrification. These samples were incubated and sampled as described above and

production of $CO_2$ during the 10 day incubation was taken as an estimate of microbial respiration. Production of $NO_3^-$ and $NH_4^+$ over the 10 day incubation was taken as an estimate of potential net N mineralization and production of $NO_3^-$ was taken as an estimate of potential net nitrification.

A denitrification enzyme assay (DEA) was used to measure the rate of potential denitrification (Smith and Tiedje, 1979; Groffman et al., 1999). Samples were placed in 125 mL Erlenmeyer flasks and amended with $NO_3^-$, glucose

as a source of C / energy, and chloramphenicol to block production of new enzymes during incubation. The flasks were sealed with rubber stoppers, flushed repeatedly with $N_2$ gas to create anaerobic conditions, amended with acetylene ($C_2H_2$) gas, and placed on a shaker at 125 rpm. The headspace of the flasks was sampled (8 mL) by



syringe after 30 minutes and 90 minutes of incubation.  Samples were analyzed for nitrous oxide ($N_2O$) with a Shimadzu GC-2014 gas chromatograph.


### 3.4 Statistical analysis

SPSS version 28 was used for all analyses.  One-way analysis of variance (ANOVA) with post hoc multiple comparison tests were run on all response variables using location (GP, HHW, CSP), texture (packed, sandy,

watery), and color (yellow, white, red, light gray, dark gray, brown) as grouping factors.  A Sidak adjustment was applied to the post hoc multiple comparison tests because the number of samples was not equal across groups. Spearman's rho was used to evaluate linear correlations because the data were not normally distributed and had a heavy positive skew.  ANOVA pretest results for homogeneity of variance were reinforced with Welch and Brown – Forsythe tests.  Kruskal Wallis analysis was run to increase confidence in results for groupings by color, texture, and

location.  Mann-Whitney analysis with the Monte Carlo option was used to further reinforce ANOVA results grouped by location.  Significance levels were evaluated based on $\alpha < 0.01$ indicating a strong statistically different significance, $\alpha < 0.05$ indicating a standard statistically different significance, and $\alpha < 0.1$ indicating a marginal statistically different significance.  Since ANOVA results yielded the same significance parameters for the majority of variables as Kruskal Wallis and Mann-Whitney tests, ANOVA values are reported in the results section.   Any

variables that yielded significant differences only from ANOVA analysis are notated.

### 4 Results

All samples had detectable amounts of total C, N, and OM (Fig. 2).  The dark gray samples had significantly higher OM, total C, total N, and C/N ratio than several of the other clay color types.  The packed clay had the highest

values of all these variables among the texture groups, and significant differences were found for OM between packed and sandy clays.  ANOVA post hoc analysis identified further significant differences in the C/N ratio when comparing dark gray clay to the Fe bearing red, yellow, and brown clays.  A number of packed, dark gray and light gray samples contained concentrations of N exceeding 1000 mg N $kg^{-1}$ which is considered ecologically significant for geologic N (Holloway and Dahlgren, 2002)**.**





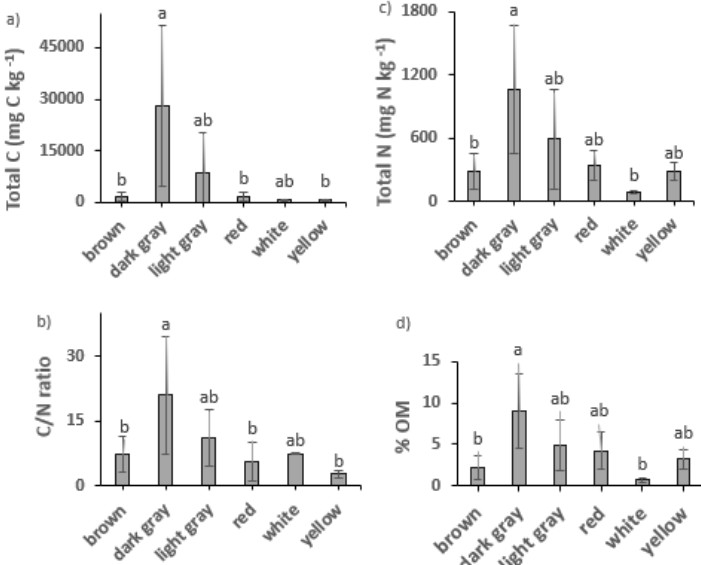


**Figure 2: Total C, N, OM content, and C/N ratio of different colored clays. Values are mean ± SD. Values with different superscripts are significantly different (p < 0.05) except: in total C dark gray clay was marginally different (p < 0.10) from yellow clay; in total N dark gray clay was marginally different (p < 0.10) from brown and white clays; C/N ratio in dark gray clay was marginally different (p < 0.10) from brown clay.**


pXRF scans yielded semi-quantitative results for selected elements of interest (Table 1).  P was highest in brown clay and in watery clay with marginal differences between GP and HHW.  Cl was highest in yellow, brown, and red clays, and in watery clays with significant differences among color groups and locations.  Ti was highest in yellow, red, and brown clays, and in packed clays with significant differences among packed and sandy clay textures and

marginal differences between GP and CSP locations.  Mn was highest in brown clays and watery clays, with significant differences among color groups, textures, and locations.  Fe was highest in brown, red, yellow clays, and in watery clays, with significant and marginal differences among color groups and locations.  Rb was highest in dark gray, and in packed clays with significant differences among color groups and locations.  Sr was similar across color groups with the highest concentrations in the light gray, dark gray, brown, and yellow samples.  Zr was highest in

brown and white samples and in sandy and watery clays, with significant and marginal differences among color groups, textures, and locations.  Zn was highest in brown and dark gray, and in watery clays, with significant and marginal differences among color groups and locations.  The Rb/Sr ratio was highest in dark gray clays, nearly the same across textures, with significant differences among color groups and locations (locations had additional marginal differences).  There were significant correlations between P and Cl, P and Mn, P and Fe, Cl and Mn, Cl

and Fe, Ti and Zr, Mn and Fe, Mn and Zn, Zn and Rb, Zn and Sr, Rb and Sr, Sr and Zr, and marginal correlations between P and Ti, and Ti and Fe.



| Garvies Point | P | Cl | Ti | Mn | Fe | Rb | Sr | Zr | Zn | Rb/Sr ratio |
|---|---|---|---|---|---|---|---|---|---|---|
| dark gray | 10991.38 | 140312.87 | 4171.20 | 67.73 | 6736.80 | 139.20 | 73.93 | 288.47 | 65.47 | 1.89 |
| light gray | 3372.50 | 133413.83 | 3441.33 | 49.33 | 4820.00 | 111.33 | 70.17 | 284.33 | 45.17 | 1.56 |
| red | 30710.67 | 161559.78 | 4884.67 | 52.33 | 27106.89 | 98.67 | 67.67 | 294.67 | 48.56 | 1.46 |
| white | 9194.00 | 126827.67 | 4351.00 | 85.67 | 6296.33 | 137.00 | 75.67 | 335.33 | 48.33 | 1.81 |
| yellow | 25721.33 | 160147.78 | 4566.11 | 54.00 | 16648.44 | 110.33 | 72.22 | 340.44 | 34.00 | 1.52 |
| **Hempstead Harbor Woods** | | | | | | | | | | |
| brown | 14576.00 | 152885.00 | 3806.00 | 204.67 | 14309.00 | 93.67 | 80.33 | 330.00 | 73.00 | 1.17 |
| light gray | 8068.67 | 123039.50 | 3832.17 | 40.17 | 6617.50 | 96.61 | 76.56 | 376.56 | 34.06 | 1.37 |
| red | 5371.50 | 139207.00 | 4649.67 | 23.33 | 7586.33 | 42.67 | 41.00 | 448.33 | 33.67 | 1.04 |
| white | 442.00 | 135305.00 | 3644.33 | 32.33 | 4786.33 | 79.67 | 51.00 | 804.67 | 23.00 | 1.56 |
| **Caumsett State Park** | | | | | | | | | | |
| brown | 31116.83 | 161317.92 | 3584.17 | 352.92 | 25723.83 | 98.08 | 72.08 | 676.67 | 74.75 | 1.39 |
| red | 3942.00 | 142768.00 | 3741.00 | 81.00 | 18065.67 | 67.33 | 62.33 | 387.00 | 59.67 | 1.08 |

Table 1: Amounts of selected elements of interest detected by pXRF analysis; values are means in ppm.

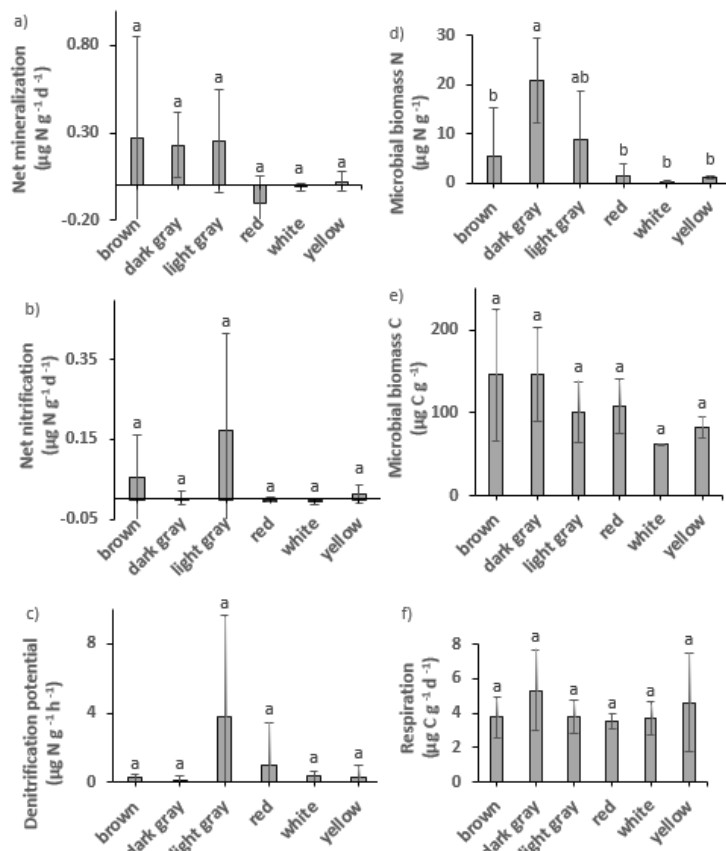


Figure 3: Microbial biomass and activity in clays classified by color: a) potential net N mineralization, b) potential net nitrification, c) denitrification potential, e) microbial biomass C, and f) respiration. Statistically different values occur in d) microbial biomass N, where values with different superscripts are significantly different at p < 0.05 except that dark gray clay was marginally different (p < 0.10) from brown and white clays. All values are mean ± SD.




Most clay types exhibited detectable amounts of microbial biomass and activity (Fig. 3). Microbial biomass C was significantly higher in the watery clay than in the packed or sandy clay. A significant difference between watery and sandy clays was identified by ANOVA post hoc analysis. There were no significant differences in microbial biomass C with clay color. Soil respiration was detectable in all materials but there were no significant differences

with clay color or texture. There were significant correlations between microbial biomass C and total N, total C (Fig. 4), OM, and respiration. Respiration was significantly correlated with total N (Fig. 4) and OM, and was also marginally correlated with total C and C/N ratio (Fig. 4).

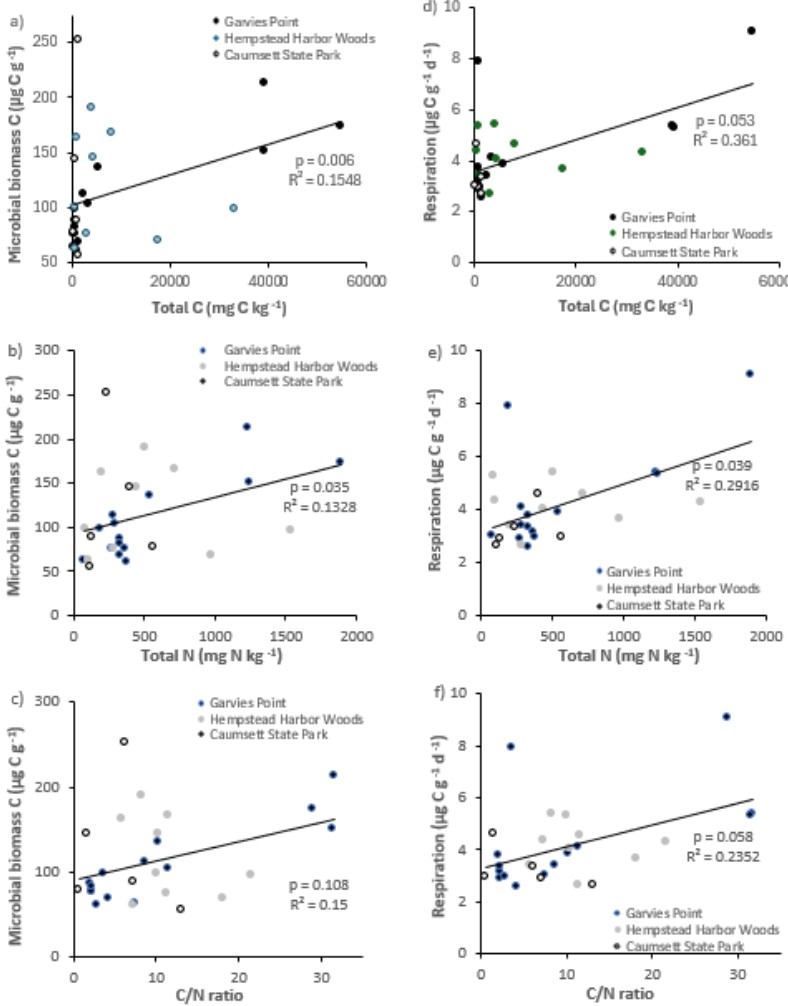

**Figure 4: Regression of microbial biomass C and respiration against total C, total N, and C/N ratio. Values are represented for each location; trendline is indicated among all locations. Significant differences occur where p < 0.05.**



There were significant differences in microbial biomass N among textures, and amounts were highest in dark gray and packed clays. We detected both potential net N mineralization and N immobilization (negative net N

mineralization) in our samples (Fig. 3). Potential net N mineralization was highest in brown and in packed clays, with significant differences among samples from all locations. Net N mineralization was marginally different between the GP and HHW locations.

Microbial biomass N was significantly correlated with total C, total N, C/N ratio (Fig. 6), potential net nitrification, OM, and TIN. Total N was significantly correlated with microbial biomass C, and respiration (Fig. 4), OM, TIN

(Fig. 5), microbial biomass N, potential net N mineralization (Fig. 6), and total C. Potential net N mineralization was significantly correlated with total C, total N, C/N ratio (Fig. 6), TIN, microbial biomass N, and potential net nitrification. Total N and the C/N ratio were marginally correlated.

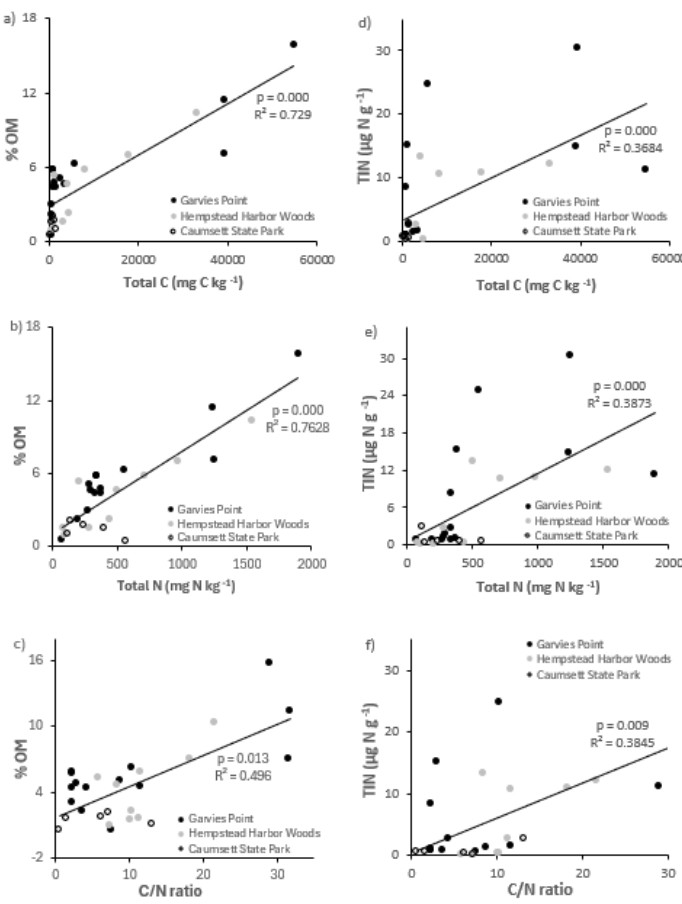

**Figure 5: Regression of OM and TIN against total C, total N, and C/N ratio. Values are represented for each location;**
**trendline is indicated among all locations. Significant differences occur where p < 0.05.**




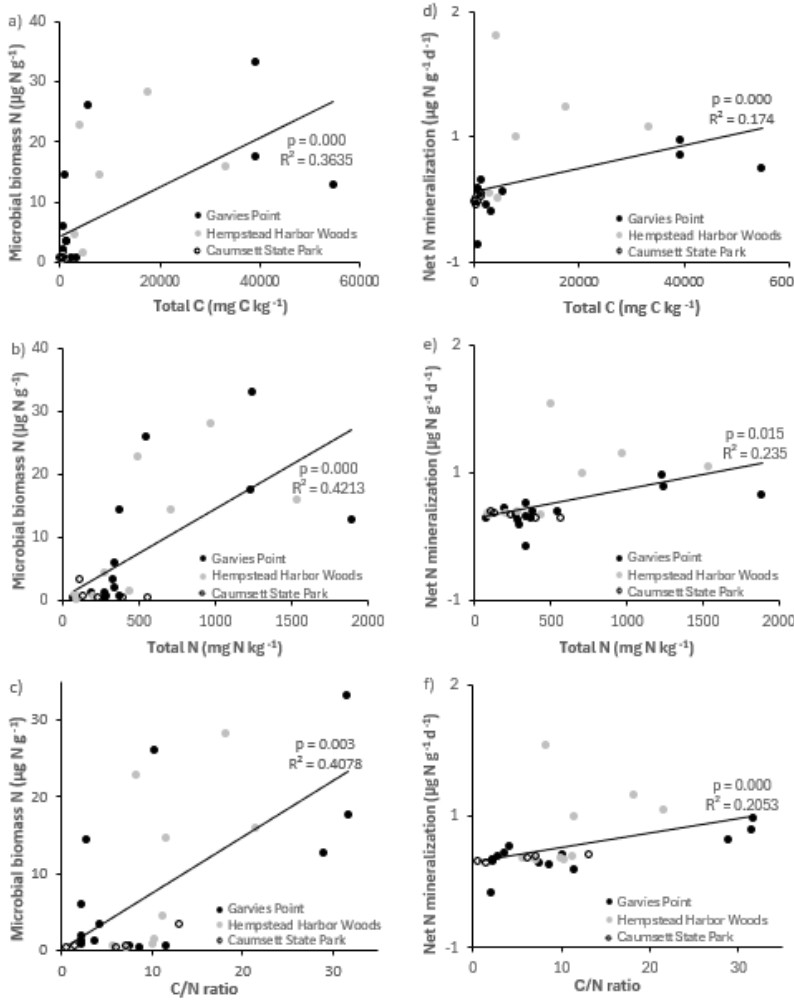

**Figure 6: Regression of microbial biomass N and net N mineralization against total C, total N, and C/N ratio. Values are represented for each location; trendline is indicated among all locations. Significant differences occur where p < 0.05.**

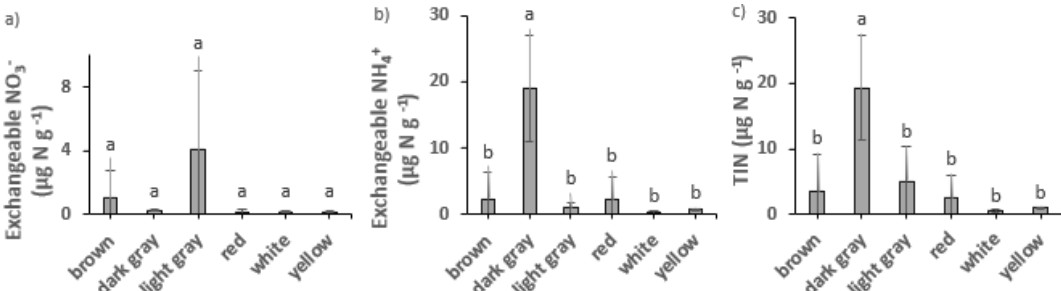

**Figure 7: Exchangeable $NO_3^-$, $NH_4^+$, and TIN in clays of different colors. Values are means ± SD. Values with different superscripts are significantly different at p < 0.05.**



The highest amounts of potential net nitrification were found in light gray (Fig. 3) and packed clays, with significant
differences between the GP and HHW locations identified by ANOVA post hoc analysis. The highest amounts of
$NO_3^-$ were found in light gray (Fig. 7) and packed clays, with significant differences between GP and HHW, and
between HHW and CSP identified by ANOVA post hoc analysis. The highest amounts of $NH_4^+$ were found in dark
gray (Fig. 7) and packed clays, with significant differences among color groups. The highest amounts of TIN were
found in dark gray and packed clays, with significant differences among color and texture groups. Kruskal Wallis
analysis identified a marginally significant difference for TIN by location. There were no significant differences in
denitrification potential.

Potential net nitrification was significantly correlated with total C, C/N ratio (Fig. 8), microbial biomass N, potential
net N mineralization, and marginally correlated with TIN. Denitrification potential was significantly correlated with
the C/N ratio (Fig. 8), suggesting joint control by C and N availability. There were significant correlations between
$NH_4^+$ pools and the following variables: total C, total N, C/N ratio (Fig. 9), microbial biomass N, potential net N
mineralization, TIN, and OM. $NO_3^-$ pools were significantly correlated with total N (Fig. 9), microbial biomass N,
and potential net N mineralization. There was a marginal correlation between $NO_3^-$ and total C (Fig. 9).

**5 Discussion**

Our study sought to explore organic-mineral interactions by investigating the functional performance of clays
through measurements of biogeochemical processes occurring within their structures. This research aimed to
expand knowledge beyond the traditional approach of investigating C and N cycle dynamics based on single
predictor values such as phyllosilicate clay, abundance of a certain mineral species, and specific surface area and
adsorption values (Kleber et al., 2021) by making a suite of measurements of microbial biomass and activity
relevant to C and N cycles. This approach allowed us to determine that ancient clays are contributing to
contemporary Critical Zone biogeochemical processes at ecosystem and landscape scales.

**5.1 N, C, and OM in geological materials**

Over geological time scales, N and C cycle from the Earth's surface to depth through subduction zones and are
returned to the surface through arc magmatism. Geological cycling of N is greatly facilitated by storage of $NH_4^+$ in
silicate minerals. $NH_4^+$ in minerals is a major influence on fluxes between reservoirs (Halama and Bebout, 2021),
thus giving geological materials such as the clays we have analyzed a potential role in ecosystem function.

For contemporary ecosystem processes, it is generally assumed that geologic materials are not an important source
of N (Schlesinger, 2013), as the dominant global pool of N is in the atmosphere. The focus of much N cycle
research is on the energetically expensive movement of atmospheric N into biological pools (Galloway et al., 2004).
The largest pools of N in ecosystems are in particulate and dissolved OM pools in soils, sediments, and the ocean
(Groffman et al., 2021). Recently, recognition of the fact that much of this organic N becomes incorporated into



geological materials has fueled interest in the role of these materials in contemporary ecosystem processes. Recent analyses show that $10^{21}$ g of global fixed N were incorporated into sedimentary rocks by the burial of OM in marine and freshwater sediments (Morford et al., 2011). These analyses have stimulated studies of the movement of deeply
buried organic N into actively cycling pools of N in soils and vegetation (Houlton et al., 2018) and interest in the exposure of ancient materials at the soil surface, such as the clays collected for this study.

At our study sites, burial of OM likely occurred in a Late Cretaceous shallow delta (Fuller, 1914). The N content of these samples is within the range of previous reports of sedimentary and metasedimentary rock N content of 200 –
1200 mg N kg$^{-1}$ (Holloway and Dahlgren, 2002; Morford et al., 2011). Organic-rich marine sediments commonly exceed 1000 mg N kg$^{-1}$ and some of our samples fit this criterion (Li, 1991; Morford et al., 2011). As these bedrock materials weather, N is released in plant available forms that stimulate ecosystem productivity and C storage (Dahlgren, 1994; Morford et al., 2011). Bedrock is also a source of N to aquifers, which is relevant for our samples which have a hydrogeologic origin from exposures of aquifer margins in a region with great concern about
groundwater N pollution (Karamouz et al., 2020).

The surface exposure of the clays in our study allowed us to directly measure microbial biomass and activities that are central to biogeochemical cycling of C and N. These measurements shed light on the role that these secondary minerals, that were produced by the weathering of primary silicate minerals which originated at depth, may be playing in the contemporary N cycle on Long Island and in the Critical Zone elsewhere on Earth. These
measurements also allowed for comparison of these geological materials with surface and subsurface soils in the region that have been assayed with the same methods (Groffman et al., 2009; Morse et al., 2014). Our analysis showed significant microbial biomass and activity in many samples, with much of the variation in activity driven by the total C and N content of the samples. The results strongly support the idea that ancient geologic materials play a role in contemporary N and C cycling in the Critical Zone.

Biogeochemical processes are influenced by microbial-mineral associations that influence the rates and magnitudes at which biogeochemical reactions occur. The structure of the mineral space influences the extent of biologically mediated reactions occurring within this habitat. Clays are expected to support less of these processes due to their tightly bound layered structure and fine particle size (Kleber et al., 2021). This is consistent with the lower amounts
of activity detected in our samples compared to other soils, discussed in section 5.2. The tight structure of the clay offers a micro-environment that is more constant over time, as compared to the more dynamic conditions found within soil (Kleber et al., 2021). The longevity of a micro-environment is affected by both OM and mineralogy, with kaolinitic 1:1 clays and oxides producing longer lasting environments (Six et al., 2000). The smaller pore size of clay facilitates anaerobic conditions leading to higher denitrification rates in clay rich materials than in sandy
ones (Li et al., 2023; Pihlatie et al., 2004). This relationship between particle size and microbial activity was evident in the analysis results obtained from our samples.



What then are the sources of the OC supporting microbial biomass and activity in the samples in this study, which varied with clay color and texture? Most C in coarse clay (0.2 – 2.0 µm) has been documented to be in the form of

charcoal or black carbon (BC) (Laird et al., 2008). We have observed BC within the clay layers at the GP site, both distinctly layered with white clay and as small inclusions in some of the gray clay. This presence indicates that BC has become incorporated into silt and clay fraction minerals in the geological unit that is exposed at the GP site, through processes such as adsorption of dissolved biogenic compounds onto the clay particle surfaces (Laird et al., 2008).


Clay color is also affected by Fe, which likely contributed to the pigmentation of the red, yellow, and brown clays in our study. Our pXRF scans detected Fe content as high as 4.55 % in some samples. Possible sources of Fe include oxidative reactions and organic ligand bonding, both of which can be catalyzed by bacteria. The Fe content of Long Island's aquifers is driven by processes that include oxidative dissolution of minerals such as pyrite ($FeS_2$) (Brown

and Schoonen, 2004). Pyrite inclusions occur in the clays at these study sites, and our pXRF scans detected the presence of S in addition to the Fe in our samples. Chelation of ions and organic acids is common in the Critical Zone. In the presence of water, Fe is one of the metal ions that can chelate with organic acids and become mobilized through the subsurface (Schroeder, 2018). Furthermore, Fe, Ti, and Mn, all of which were detected in our samples, play a key role in mineral and organic oxidation reactions. Fe(II), $FeO_2$, and $TiO_2$ bearing clays produce

the highest amounts of reactive O species, which in turn react with organic C to transform soil and sediment OM, and to produce $CO_2$ as the end-product of organic C oxidation (Kleber et al., 2021). Mn oxides are the strongest naturally occurring oxidants and play an important role in organic C transformation (Remucal and Ginder-Vogel, 2014).

The much higher occurrence of significant differences among the clays when grouped by color and texture, as

opposed to when grouped by location, indicates that land use history and other local characteristics have less or no effect, and that the main differentiator for microbial activity is the clay material itself and variations therein, such as Fe oxides, trace elements, and variations in clay speciation. It also indicates that the clays from the different locations are likely part of the same larger formation (presumably Raritan or Magothy).

**5.2 Are these ancient clays contributing to contemporary biogeochemical processes?**

Our study was driven by three questions, the first of which was the most fundamental: do these materials support living microbial biomass? Our analysis detected a living microbial community on most of the clay samples. There were strong positive relationships between total C content, total N content, and microbial biomass and activity. The

dark gray clays had the highest total N, total C, and microbial biomass N, and microbial biomass C was highest in both dark gray and brown clays. White clays had the lowest contents of these variables, indicating that inclusions of Fe and C (discussed in section 5.1) in the clays play a role in their capacity to sequester C and N and support microbial biomass and activity.




While our clay samples had significant amounts of microbial biomass and activity, they were less "alive" than surface soils that have been studied across the northeastern U.S. Both surface organic and subsurface mineral horizons of fully developed soils often have substantial C and N pools (Bohlen et al., 2001), with variations in distribution. We compared the data of our clay samples to two studies that used the same analysis methods (Groffman et al., 2009; Morse et al., 2014) (Table 2), to elucidate the differences in microbial biomass and activity

between different soils and the clays. In a comparison with forest, agricultural, and grassland soils in the Baltimore, Maryland metropolitan area, respiration and microbial biomass C of the clays (respiration 4.1 µg C g$^{-1}$ d$^{-1}$; microbial biomass C 114 µg C g$^{-1}$; ratio 28) was closest to the agricultural soil (respiration 6.7 mg C kg$^{-1}$ d$^{-1}$; microbial biomass C 224 mg C kg$^{-1}$; ratio 33) (Groffman et al., 2009) and lower than the forest or grassland soils.

We further compared our results to soils in various spodic hydropedologic settings in a northern hardwood forest at the Hubbard Brook Experimental Forest, New Hampshire (Morse et al., 2014). These settings included typical podzols (T), bimodal podzols (Bi), Bh podzols (Bh), and seeps, and three depths (Oi/Oe, Oa/A; B horizons) were sampled. Common features of spodsols are complexation with Al and Fe (often organic complexation) and enrichment with Fe and Mn (Van Ranst et al., 2018), thus sharing some elemental features with our clay samples

since Al is a major constituent of clay and pXRF scans confirmed the presence of Mn and Fe in our samples.

| | | Total C | Respiration | MBC | MBN | NH$_4^+$ | NO$_3^-$ | TIN | ratio of MBC/Total C | ratio of MBC/Respiration |
|---|---|---|---|---|---|---|---|---|---|---|
| Groffman et al., 2009 | | | mg C kg⁻¹d⁻¹ | MBC mg C kg⁻¹ | | NH$_4^+$ mg N kg⁻¹ | NO$_3^-$ mg N kg⁻¹ | mg N kg⁻¹ | | |
| Forest mean | | | 10.3 | 346 | | 2.1 | 0.4 | 2.5 | | 33.592 |
| Agriculture mean | | | 6.7 | 224 | | 0.9 | 8.7 | 9.6 | | 33.433 |
| Grassland mean | | | 8.2 | 306 | | 0.5 | 1.2 | 1.7 | | 37.317 |
| Values source: Table 3 | | | | | | | | | | |
| | | | | | | | | | | |
| Morse et al., 2014 | | | | | | | | | | |
| Hydropedologic setting | Horizon | Total C mg C kg⁻¹ | | MBC ug C g⁻¹ | MBN ug N g⁻¹ | NH$_4^+$ ug N g⁻¹ | NO$_3^-$ ug N g⁻¹ | TIN ug N g⁻¹ | ratio of MBC/Total C | |
| T: typical podzols | Oi/Oe | 506000 | | 4780 | 774 | 152 | 16.8 | 168.8 | 0.009 | |
| | Oa/A | 329000 | | 2320 | 320 | 30.4 | 18.3 | 48.7 | 0.007 | |
| | mean of surface Oi/Oe, Oa/A | 417500 | | 3550 | 547 | 91.2 | 17.55 | 108.75 | 0.008 | |
| | B > 10 cm | 74000 | | 481 | 32.4 | 2.4 | 3.33 | 5.73 | 0.007 | |
| Bi: bimodal podzols | Oi/Oe | 378000 | | 6210 | 511 | 83.7 | 9.6 | 93.3 | 0.016 | |
| | Oa/A | 158000 | | 1300 | 226 | 6.7 | 9.97 | 16.67 | 0.008 | |
| | mean of surface Oi/Oe, Oa/A | 268000 | | 3755 | 368.5 | 45.2 | 9.785 | 54.985 | 0.012 | |
| | B > 10 cm | 56000 | | 275 | 21.6 | 2.12 | 0.53 | 2.65 | 0.005 | |
| Bh: Bh podzols | Oi/Oe | 443000 | | 8230 | 710 | 144 | 17.8 | 161.8 | 0.019 | |
| | Oa/A | 234000 | | 3310 | 333 | 4.93 | 15.9 | 20.83 | 0.014 | |
| | mean of surface Oi/Oe, Oa/A | 338500 | | 5770 | 521.5 | 74.465 | 16.85 | 91.315 | 0.016 | |
| | B > 10 cm | 60000 | | 569 | 36 | 1.46 | 2.67 | 4.13 | 0.009 | |
| Seep | Oa/A | 232000 | | 5060 | 238 | 3.18 | 1.02 | 10.2 | 0.022 | |
| Values source: Table 1 | | | | | | | | | | |
| | | | | | | | | | | |
| This study | | Total C mg C kg⁻¹ | ug C g⁻¹ d⁻¹ | MBC ug C g⁻¹ | MBN ug N g⁻¹ | NH$_4^+$ ug N g⁻¹ | NO$_3^-$ ug N g⁻¹ | TIN ug N g⁻¹ | ratio of MBC/Total C | ratio of MBC/Respiration |
| All locations, surface, mean | | 8220.18 | 4.1 | 113.53 | 7.64 | 4.63 | 1.47 | 6.11 | 0.014 | 27.690 |

HPS: hydropedologic setting, T: typical podzols, Bi: bimodal podzols, and Bh: Bh podzols; MBC: microbial biomass C, MBN: microbial biomass N.

**Table 2: comparison of response variables from this study to data from Groffman et al., 2009 and Morse et al., 2014. All three studies used the same sample analysis. Respiration and microbial biomass C of the clays in this study was closest to agricultural soils in the Baltimore, Maryland area (Groffman et al., 2009), and most comparable to the B horizon of the Bi**
**setting at Hubbard Brook Experimental Forest, New Hampshire (Morse et al., 2014).**

Our samples, all of which were collected from the surface, were most comparable to the B horizons and to the Bi setting from Morse et al. (2014) (Table 2). B horizons in the Bi setting soils had the lowest total C and microbial biomass C and N contents of the Hubbard Brook soils but were still much higher than our clay values. The clay had



a higher microbial biomass C / total C ratio (0.014) than the surface (0.012) and B (0.005) horizons at Hubbard
Brook suggesting relatively high OM quality in the clays.

Morse et al., (2014), found that differences in total C and N content were more notable across soil horizons than
across hydropedologic setting, where the B horizon, which accumulated Al and Fe, had less C and N and microbial

biomass and activity.  The dominance of chemical (OM quality) versus location controls is consistent with the
stronger differences that we observed with color than with location.  The composition of OM in clay fractions differs
from that in sand and silt fractions, and there is usually a decrease in C/N ratio as particle size decreases from coarse
silt to fine clay (Laird et al., 2008).  In clay, the packed textures dominated by smaller particle size contain more OM
than larger particle sandy samples, but the C/N ratio increases with smaller particle size.  In our samples, dark gray

clays had the highest total N, total C, and microbial biomass N; microbial biomass C was highest in both dark gray
and brown clays.  White clays had the lowest contents of these variables, indicating that inclusions (such as Fe, BC,
and others) in the clays play a role in increasing their capacity to accommodate C and N rich contents.

The second question our study addressed is whether the clays support an active N cycle.  We measured several

indices of N cycling activity that showed that the exposed clays do indeed support and active N cycle and have
potential to supply N to support plant growth.  Microbial biomass N is an index of the size of the actively cycling
labile N pool in soil.  Mineralization, the production of simple, soluble, inorganic forms of N that are a dominant
source of N for plant growth, is strongly tied to the C cycle (Hart et al., 1994).  When microbes degrade N
containing compounds during mineralization, their N is converted to proteins which release ammonia ($NH_3$) that

converts to $NH_4^+$, and their C is converted to biomass or $CO_2$ (Groffman et al., 2021).

Microbial biomass N was highest in dark gray and packed clays, and correlated with potential mineralization and
nitrification, as well as with $NH_4^+$, $NO_3^-$, TIN, and total N.  Potential net N mineralization was highest and
comparable in brown and light gray and dark gray clays, and lowest in yellow clays.  Microbial biomass N, potential

mineralization and nitrification, $NH_4^+$, TIN, and total N were all strongly correlated with total C, and $NO_3^-$ was
marginally correlated with total C, indicating C content as a driver in the coupling of these processes in the clays.
Microbial biomass N, potential mineralization, and nitrification were also correlated with OM.  Respiration had a
stronger correlation with total N than with total C, but the strongest correlation was with microbial biomass C, and
there was no correlation between respiration and microbial biomass N.  The C/N ratio was also highest in dark gray

and packed clays and lowest in Fe rich red and yellow, and in watery clays.

Our findings have general similarities to those of Morse et al. (2014), who found that clay rich B horizons have
lower rates of biogeochemical activity (lower net N mineralization and net nitrification potentials) as well as smaller
C and N pools than surface soils with less clay and Al.  Clay minerals play a role in the C and N adsorption and

stabilization in soil.  The storage potential of C is influenced by the size of the silicate mineral's surface area and the
amount of cations adhering to these minerals (Kahle et al., 2002).  Adsorption rates and amounts of OM on mineral



surfaces are influenced by variations in aqueous solution dynamics, mineralogy, and OM chemistry, while OM affects mineral growth, transformation, and dissolution (Kleber et al., 2021).

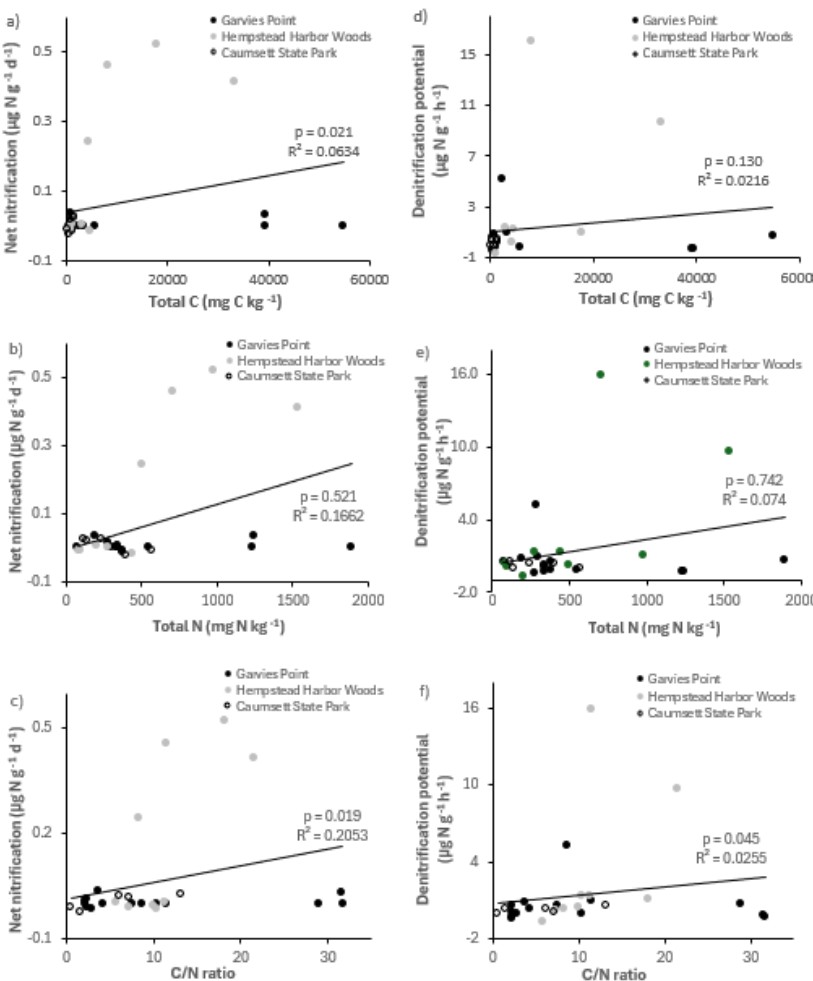


**Figure 8: Regression of net nitrification and denitrification potential against total C, total N, and C/N ratio. Values are represented for each location; trendline is indicated among all locations. Significant differences occur where p < 0.05.**

The third question we asked is whether the clays are a source of N pollution.  There is great concern about N

pollution of groundwater and coastal waters in our region of study (Karamouz et al., 2020).  There is particular

concern about  $NO_3^-$, the most highly mobile form of reactive N that is a drinking water pollutant and a prime cause

of eutrophication in coastal waters (Conley et al., 2009).  We therefore assessed the potential of these clays to

contribute to high levels of $NO_3^-$ in the environment by measuring both $NO_3^-$ pools as well as processes that produce

(nitrification) and consume (denitrification) $NO_3^-$.






Nitrification is carried out by chemoautotrophic bacteria that oxidize $NH_4^+$ into $NO_2^-$ which is further oxidized into $NO_3^-$. When this process is stimulated by the addition of N through application of fertilizers, atmospheric deposition, groundwater and runoff sources, it can lead to excessive production of $NO_3^-$ (Groffman et al., 2021). Denitrification is an anaerobic process that converts $NO_3^-$ to gaseous forms NO, $N_2O$, $N_2$ (Robertson and Groffman,

2015), removing reactive N from the soil and facilitating the cycling of N between the biosphere and lithosphere to the atmosphere.

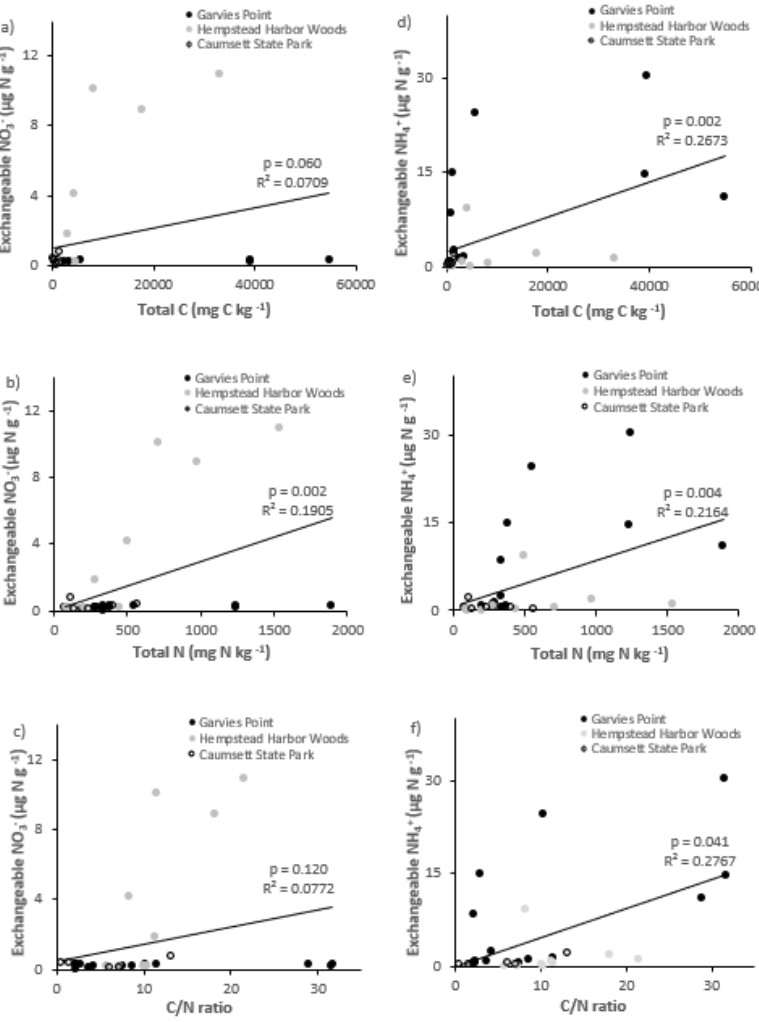

**Figure 9: Regression of exchangeable $NO_3^-$ and $NH_4^+$ against total C, total N, and C/N ratio. Values are represented for**
**each location; trendline is indicated among all locations. Significant differences occur where $p < 0.05$.**



### 5.3 N Pollution


N pollution occurs in the air, soil, and water. In the air it is the addition of volatile forms of N gasses, such as NOy caused by industrial and commercial activity; in soil there is often an overload of nutrients from overuse of fertilizers and deposition from the atmosphere; in water a major cause is sewage system drainage and agricultural runoff, causing eutrophication which leads to dead zones in the oceans, in the most extreme scenario. The main

source for N pollution (in the form of $NO_3^-$) to Long Island's aquifers is from septic systems, with additional inputs from agriculture, lawn care, and atmospheric deposition (Szymczycha et al., 2017). $NO_3^-$ fluxes in Long Island's aquifers can negatively impact coastal ecosystems (Karamouz et al., 2020) and drinking water quality. The clay units included in Long Island's aquifer and aquitard strata can act as confining units, and as sinks or sources of N pollution.

The clays examined in this study could act as a minor source or a potentially significant sink for N pollution in an aquifer system. Clays with significant rates of potential net nitrification (especially the light gray clay) could be a source of $NO_3^-$ if they were in contact with the aquifer. However, rates of denitrification potential were generally much higher than potential net nitrification, suggesting that these clay materials are more likely to act as $NO_3^-$ sinks. The light gray clays have the potential to act as both $NO_3^-$ sources and sinks depending on environmental conditions,

while the white and Fe-bearing (red, yellow) clays have the potential to act as $NO_3^-$ sinks.

### 6 Conclusions

This investigation shed light on the amount and type of microbial activity that occurs in geological microhabitats in the Critical Zone at a coastal exposure of temperate northern latitude and allowed us to evaluate their potential local

and regional impact.

Our analyses have taken a step towards better understanding the nature and extent of the biogeochemical activity that these types of microhabitats support. The approach of using laboratory measurements of microbial biomass and activity in ancient materials was successful in characterizing the biogeochemical potential of these materials, even at low levels, and could be applied in other Critical Zone studies.

The results from this study provide quantitative data showing microbial activity of silt and clay fraction materials of hydrogeologic origin, and confirm that these materials contain ecologically significant concentrations of geologic N exceeding 1000 mg N kg$^{-1}$ (Holloway and Dahlgren, 2002). Their surface exposure allowed us to explore the interaction between ancient geological stratigraphic components and modern day environmental conditions. This type of interaction is illustrative of the conceptual reach of Critical Zone science and the cohesive understanding of

multiple differing factors that it provides. The unity of various disciplines and their individual approaches to



investigation allow for greater understanding of the ensuing conditions that occur in the Critical Zone, where the resulting processes work together to support all living organisms.

Our results advance the emerging science of the geological N cycle and clearly show that ancient geological materials are contributing to contemporary biogeochemical processes in the Critical Zone of our study region.

Further, we have shown that these materials support a wide range of N cycle processes encompassing mineralization, immobilization, nitrification, and denitrification. There is a clear need for further research on how clay physical and chemical characteristics influence the flows of N to and from clays and biogeochemical processes in the Critical Zone, and to see how these materials and processes are contributing to the growth of vegetation and the dynamics of pollutants in the Critical Zone of this dynamic, densely populated, and environmentally sensitive

region.

**Data availability**

The supplement to this article is available at doi: 10.5281/zenodo.12702768


**Author contributions**

VMA conceptualized the study, performed field work and sample collection, laboratory analyses, data analysis, and wrote the manuscript; PMG designed the study, provided laboratory methodology, access, and supplies, interpreted results, co-wrote, and revised the manuscript; ZC provided laboratory equipment and access, interpreted results, and

revised and edited the manuscript; DES interpreted results, revised and edited the manuscript.

**Competing interests**

None of the authors have any competing interests.

**Acknowledgements**

The authors thank Clare Kohler and Kaitlin McLaughlin for help with laboratory analyses; and Veronica Natale and Dr. Herbert Mills for help obtaining samples.



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
