# Peer review of "Ancient clays support contemporary biogeochemical activity in the Critical Zone"

_EGUsphere, 2024_

## Author Comment (AC1)

**Response to Reviewer Comments (Egusphere-2024-1165)**

We thank the reviewers and the editor for their valuable comments and insights. We have addressed each comment as described below. The vast majority of reviewer suggestions have been taken and we hope that the manuscript is greatly improved as a result. The main changes that have been made are clarification of 1) just what was sampled, i.e., geological materials and not soils and 2) our research questions, objectives and hypotheses.

NOTE: please note that the line numbers stated by the reviewers correspond to the preprint version of the manuscript, and that the line numbers stated in the author's responses correspond to the updated version of the manuscript as lines have shifted during editing.

**Reviewer #1**

The article "Ancient Clays Support Contemporary Biogeochemical Activity in the Critical Zone" by Alfonso et al. examines the interaction of Late Cretaceous clays on Long Island, NY, with modern biogeochemical processes, aiming to answer three central questions: whether these clays support microbial life, their role in nitrogen (N) cycling, and whether they function as sources or sinks of nitrogen pollution. The authors report that these clays support active microbial communities involved in carbon (C) and nitrogen (N) cycling, with findings suggesting they act more as N sinks than sources in the current landscape.

While these clay outcrops are an interesting research system, this manuscript contains several key issues that impact its clarity and effectiveness. First, it seems that there is confusion between the role of nitrogen in clays (which are part of the soil) and geologic, lithified clays (sedimentary rock). One of the main conclusions these authors present is that the nitrogen in their samples was of geologic origin. This could be true if their study system was lithified clays (sedimentary rock), but it seems that the authors sampled soil clay that is exposed to the atmosphere and biosphere. For this reason, it is more plausible that this nitrogen originates from biological inputs, such as organic debris or nitrogen-fixing bacteria. Emphasizing this point would enhance the clarity and scientific accuracy of the findings.

We have clarified (lines 179-181) that "samples were collected from unconsolidated bulk clay units that showed no evidence of modern soil development, i.e., accumulation of recent organic matter, development of horizons."

We have added some text to the Discussion (lines 372-375) that acknowledges that "while it is reasonable to assume that some of the N in our samples could be of recent biological origin and is adsorbed onto the surface of the clay particles, our sampling approach that targeted areas with no evidence of modern organic matter or soil development suggest that our analysis was of older N associated with geologic materials."

Second, there seems to be a misalignment between the research questions and the analyses. While the study addresses relevant biogeochemical questions, some analyses—such as comparisons between clay color groups—appear to lack a strong initial hypothesis or rationale. If differences in clay color and texture are central to understanding microbial and nitrogen dynamics, introducing these factors within the research questions, along with hypotheses on their significance, would clarify the study's objectives and improve reader engagement.

We have clarified that while our sampling facilitated comparison of these obvious differences, we did not have any a priori hypotheses about their causes and biogeochemical effects. Rather, our focus was the three research questions listed in the first paragraph of the paper. We do discuss (lines 421 – 445) the factors driving these differences, e.g., iron and organic matter but our analysis was not hypothesis-driven.

Further, some clay color differences are likely indicative of oxidative or reducing conditions (e.g., red-yellow-brown clays indicating oxidative conditions, dark gray-light gray-white clays suggesting reducing conditions). These environments have significant implications for microbial activity, organic matter decomposition, and nitrogen cycling. Including a discussion of these environmental differences in the context of microbial and nitrogen dynamics would enrich the interpretation of results.

We have added text (lines 438-439) to acknowledge that "these color variations indicating variation in oxidative or reducing conditions may reflect biogeochemical conditions and activity during their formation that have legacy effects on contemporary activity." The beginning of that paragraph (lines 426-430) also provides context for oxidative conditions with citations.

This manuscript would also be improved by reframing the question of N pollution. While the study explores whether clays may act as sources of N pollution, pollution often implies anthropogenic contamination. Reframing this question as "Are these clays sources or sinks of reactive nitrogen?" would allow for a more relevant exploration of nitrogen fluxes without presupposing anthropogenic contamination.

We have revised this question to read, "Are these clays a potential non-anthropogenic source of reactive N pollution in the contemporary landscape?"

Major comments:

Background: The background section provides some useful information but could be condensed for clarity. For instance, the detailed descriptions of clay mineralogy and their role in biogeochemical cycling could be distilled into a few key sentences relevant to the study's objectives. Simplifying this section would help maintain focus on the study's aims. Throughout the manuscript, the authors may consider cutting down on more textbook-like definitions.

We have greatly shortened the "clay mineralogy and biogeochemical cycling background" as suggested. Also, the background section has been reorganized for a more streamlined reading experience that we hope provides enough background for an interdisciplinary audience without being too simplistic.

Introduction: Key points from the background could be integrated here to streamline the manuscript. Additionally, a brief summary of the research approach at the end of the introduction would provide readers with a helpful roadmap. Most importantly, the authors should provide the rationale and motivation for their analytic approach.

We have added a brief summary of the research approach to the beginning of the final paragraph of the Introduction as suggested. This paragraph provides rationale and motivation for each of the

variables that we measured, i.e., microbial biomass N, mineralization, nitrification, denitrification. The background section has been streamlined by adding sub-sections.

Methods: This section is well-detailed but would benefit from visual aids (e.g., images of the clays in situ) to help readers visualize the sampling locations and types of clays. Including sample numbers, sample sizes per site and clay type, and explanations of the statistical methods would enhance reproducibility and interpretability. Specific points include:

- Clarify sample numbers, types, and reasoning behind using one-way ANOVAs over two-way ANOVAs. Please see detailed responses below.

- Explain the rationale behind grouping clays by color and texture, and clarify whether color/texture was hypothesized to impact microbial and nitrogen dynamics. We have clarified that clays were distinguished by obvious visual criteria and that we did not have any a priori hypotheses about their causes and biogeochemical effects.

- The statistical methods would be clearer if directly tied to the research questions. Including $R^2$ values for correlations would allow readers to gauge model strength and clarify interpretations. For the pairwise correlations, it seems that most of these data are log-distributed, so log transformations would be appropriate. Log transformations are typical of these types of ,data. All correlation graphs include $R^2$ values. There is also a chart in the supplemental data (table S1) that lists all the actual correlation values. Spearman's rho was used to evaluate linear correlations because the data were not normally distributed and had a heavy positive skew. ANOVA pretest results for homogeneity of variance were reinforced with Welch and Brown – Forsythe tests.  Kruskal Wallis analysis was run to increase confidence in results for groupings by color, texture, and location.  Mann-Whitney analysis with the Monte Carlo option was used to further reinforce ANOVA results grouped by location.

- What is the sample size? How many samples were retrieved from each site, of each color, and of each texture? This has been clarified in the Sampling section (lines165-167) and details are included in the supplemental materials (tables S2, S3, S4).

- Without a clear description of the research questions, it is difficult to provide feedback on the choices made in the statistical analysis. For example – why one-way ANOVAs and not two-way ANOVAs? Why is there no test for interactive effects of site, color, or texture? The research questions, which are exploratory and listed in the abstract and introduction and revisited in the results and discussion sections, do not fundamentally depend on the statistical analysis. The statistical analyses supported these questions by comparing sampling locations and color types. Decisions about particular tests were made to accommodate unequal sample groups and non-homogeneity of variance.

- What was the motivation for fitting linear correlations between response variables? What research questions and/or hypotheses are addressed here? The correlations were used to shed light on how elemental content relates to and affects biogeochemical cycling of C and N. Spearman's rho was used to evaluate linear correlations because the data were not normally distributed and had a heavy positive skew.

Results: Structure the results around mechanistic hypotheses rather than descriptive correlations. Rather than listing observed correlations, frame findings in terms of initial hypotheses about microbial activity and nitrogen cycling within oxidative and reducing conditions. We have added sub-headings to the Results section that structure the presentation around the three driving research questions of the study.

Effect Sizes and Interpretation: Including effect sizes along with significance testing will give readers a clearer understanding of the practical implications of differences. Emphasize interpretation, especially where findings may contrast or align with known behaviors in similar environmental settings. We have not added effect sizes to the significance testing. This would not be very meaningful given the unequal sample sizes and non-homogeneity of variance. More fundamentally, this is an exploratory, descriptive study not intended to produce practical implications or recommendations.

Discussion:

Relevance of N in Sedimentary Rock (Section 5.1): While the discussion on sedimentary rock N is informative, it could be more focused on the study's core material—clays—rather than extending to other geologic nitrogen sources. We have clarified (lines 392-393) that "the limited literature on the N content of geological materials does not include any studies of clay materials similar to those studied here." We did find one paper on nitrogen in illite/smectite mixed layer clays from shale which is not relevant to our study.

Interpretation of Oxidative/Reducing Conditions: Discuss the potential implications of Fe and organic matter for microbial activity, and consider that the high OM content in dark gray clays could be due to reducing conditions that inhibit decomposition. This hypothesis could explain the enhanced microbial activity in certain clays and provide a more cohesive interpretation. We have added a sentence (lines 438-439) to note that "these color variations indicating variation in oxidative or reducing conditions may reflect biogeochemical conditions and activity during their formation that have legacy effects on contemporary activity." Also, we discuss the origin of these variations in several places: lines 140-148; 420; etc.

Nitrogen Source (Line 526): The study cannot conclusively determine that the N in clays is of geological origin; it's more plausible that organic matter adsorbed onto clay particles contributes to the nitrogen pool. Acknowledging this distinction would align the study with current understanding. As noted above, we have added some text (lines 373-374) stating that "while it is reasonable to assume that some of the N in our samples could be of modern biological origin and is adsorbed onto the surface of the clay particles, our sampling approach that targeted areas with no evidence of modern organic matter or soil development suggest that our analysis was of older N associated with geologic materials.

Comparisons with Other Studies (Lines 410–425): The relevance of comparing these results to agricultural, forest, and grassland soils is unclear without a stronger argument for its relevance. The authors should provide context for this comparison, explaining why it is included and what conclusions they hope to draw from it. Without a clear rationale, including an entire table of results from the Hubbard Brook soils study seems tangential. We have clarified (lines 456-460) that "it is important to compare our results with other materials and soils in our region (the northeastern U.S.)

to evaluate just how important this activity might be." The type of geological material analyzed in this study (clay) has not been previously analyzed using these soil biogeochemical protocols. Data from this study was compared to those obtained by analyzing different geological materials (soils from differing regimes) that were analyzed in the same lab using the same methods. The variations, but even more so the similarities between the data sets provide important context for the data presented here.

Summary of Results (Lines 455–465): Some portions of this section simply summarize results without interpretation. Offering an interpretation of the findings (e.g., the significance of high OM content in dark gray clays due to potential reducing environments) would provide more depth. We have deleted this paragraph (lines 502-510 / formerly 455-465) as it does indeed simply summarize results that have already been presented. The paragraph above (lines 494-500) offers an interpretation of the finds and the paragraph below compares these new findings with previous results.

Minor comments:

Lines 45-48: Rephrase Morford et al. (2011) as an influential study rather than recent. Additionally, clarify whether the study examined soil or lithified clays. Removed the word recent as suggested (line50). Clarified that it's neither soil nor lithified, but rather unconsolidated bulk clay.

Line 118: If both clay and shale are being sampled, clearly differentiate between these materials as they may interact differently with microbial processes. We have clarified (lines 133 – 134) that the layered shale was not sampled for this study.

Lines 125-134 Seem to belong in the discussion section? Also, please describe what pXRF is. E.g. "Analysis by a portable X-ray fluorescence device (pXRF) revealed…" We mention previous studies in the introduction in order to lay the foundation as to why we are studying these clays in this analytical manner. Lines 128-136 / new lines 140-148 explain why we grouped clays by color, texture, etc and offers analytical results as justification. This section describing the color variation of the clays pertains to the material's background rather than our findings. pXRF is defined on line 144, the first time it occurs in the text.

Line 140 – It says here there are four types of clays, but the next line reads that 5 types of clays were sampled. Clarification? This was corrected on line 154. There were 5 types of clays.

Line 149 –Particle size distribution? Was this measured in this study or previously? If this is from previous research, it should be cited. If this is from this present study's analysis, these are results and should be in the results section. There is no mention of particle size on line 149. Does the reviewer refer to line 175 (now line 194) perhaps? If so, the particle size is an observational assessment of the sample.

Line 183: Replace "ignited" with "combusted" in describing the LOI process. Changed as suggested. Line 200.

Line 201: Provide a citation or rationale for using the proportionality constant of 0.41 in microbial biomass calculations. Citation added. Line 219.

Line 205 – Was nitrate and ammonium extracted with KCl before and after the incubation time? If so, this needs to be stated. The extractions are stated starting on line 218 and on line 216 in the Methods section.

Line 209 – What mass of sample did you use? What volume of nitrate solution, glucose, and chloramphenicol did you use? These details were added to the Methods section lines 226-229.

Line 230 – Meaning unclear. This sentence has been deleted.

Line 246: For pXRF analysis, define "semi-quantitative" and specify the elements of interest, along with their relevance to the research questions. Including the purpose of investigating pairwise correlations would improve focus. We have clarified (line 268) that "semi-quantitative" analysis by pXRF produces estimates rather than quantitative characterization of concentrations. The elements of interest are listed in Table 1 which is cited in the sentence. We have added detail on the potential importance of each element of interest in lines 269-284. There is also a discussion regarding elements of interest on lines 426-439.

Line 272 – It would be helpful to discuss the magnitude and direction of the differences between groups, not just that the differences were significant. We have added indicators of the magnitude and direction of differences between groups in several places in the Results section. Line 298, 337-344.

Table 1 – Sample size? Standard deviations? We added sample size for each site under Sampling in the Methods section and all samples for each site are listed in the supplemental materials. The caption was adjusted to clarify that the values are estimates. We added Table 1b of the standard deviations above line 290.

Figure 1 – Use a consistent color palette to represent sites. All sites are represented in red on the map.

Line 304 –Magnitudes and directions of these differences? Percentages were added to distinguish higher values (lines 338-344). Individual values are included in the supplemental materials.

Line 311 - Note that, while correlations are significant, small $R^2$ values suggest weak relationships. Consider tempering strong interpretations. We provide both the significance and $R^2$ values so that the reader can decide how important the relationships are.

Lines 373-380: Suggesting black carbon as the primary source of organic C may not be fully supported unless a history of fire is known for the study site. The authors here cite a study on prairie soil carbon, which is more likely fire-derived. Acknowledging that local plant material may also contribute to organic C would provide a more balanced interpretation. We have clarified (lines 417-425) that while black carbon is one possible source of OM, we are not suggesting that it is the primary source.

Lines 450-454: This section presents textbook-like information that could be simplified or distilled. Emphasize relevance to the study's aims. We think this text (starting line 495) provides useful context for an interdisciplinary audience about just what our biogeochemical measurements mean. We note that the paragraph beneath this one has been deleted.

**Citation**: https://doi.org/10.5194/egusphere-2024-1165-RC1

**Reviewer #2**

**RC2**: 'Comment on egusphere-2024-1165', Anonymous Referee #2, 08 Dec 2024

I read with interest the manuscript by Alfonso et al. on the role of ancient clays on biogeochemistry.

The questions posed on the abstract hit a bit abruptly. Just because the parent material is older why might there not be microbial activity? The questions regarding N pollution were more interesting but without context made it difficult to ascertain its importance (is there the notion that these soils are contributing to excess N on Long Island? Will this be addressed later?). We have revised the first sentence of the Abstract to include, "ancient geological materials that that are often assumed to not be biologically active."

Along these lines the arguments in the paragraph ending line 49 strike me as a bit incomplete. Is there something unique about these parent materials other than age that might make them unique, biogeochemically? Is there evidence from other older and/or not exposed (or exposed) clays that would lead one to believe that their biogeochemical cycling is different? I'm personally interested in most things biogeochemical and find value in such studies in and of themselves, but for a more general audience what should be expected to be found given previous results, and what might one learn?

We have added text (lines 40-44) to highlight that "these materials are often assumed to have little or no biogeochemical activity as they have not been involved in active biological activity for long periods of time and may lack active microbial communities and labile pools of carbon (C) and nitrogen (N) that drive this activity. We have highlighted (lines 46-48) that "Major questions center on the ability of ancient materials to support biogeochemical processes related to the cycling of C and N that underlie plant and microbial activity, which underlies environmental and ecosystem "services" of interest to society. "

Reading on to the background section, it may help to merge the background and introduction to give people context first before going into the details; I find that the exposition does not funnel well from broader importance to particular study. The Background section itself doesn't necessarily flow with the introduction; it could almost be a supplement or a sidebar. The geological history is critical of course, but restructuring is required and after reading it a couple of times it needs a re-write (if pathways is the noun, progresses shouldn't be the verb).

The Introduction and Background sections have been extensively revised and restructured as suggested by both reviewers. The writing has been checked, e.g., pathways is now linked to progress as a verb.

(Mason should be capitalized because it is a formal name of the founder of this type of jar.) This correction has been made.

The typesetters will likely catch this but the digital quality of figures 2 & 3, actually most figures, should be improved. The digital quality has been previously approved during the technical review phase.

The importance of the analysis by color of clay really needs to be described sooner. Of course these clays are different, but for a reader who is not an expert in clays, what should one expect to find and how does this study build on that knowledge? As described above in our responses to reviewer #1, we have clarified that while our sampling facilitated comparison of visually obvious differences in color, we did not have any a priori hypotheses about their causes and biogeochemical effects. Rather, our focus was the three research questions listed in the first paragraph of the paper. We do discuss (lines 405-434) the factors driving these differences, e.g., iron and organic matter but our analysis was not hypothesis-driven.

292: avoid using 'marginally correlated'. Pick a significance level and stick with it. This can also help streamline the results section. Criteria for significant and marginal differences are explained in detail in the Statistical Analysis section (lines 246-248). We opted to include marginal differences in order to provide a broader picture of relationships occurring in natural samples that exhibit a great deal of variation in their features and content. We find that this provides the audience with a better understanding of the biogeochemical activity and its relationship to mineralogy in this system.

Note in the figures that p can't technically be '0.000'. Use either < 0.05 or less than some other small value. We have changed 0.000 to $p < 0.001$.

For context, in case I'm seeming obtuse, using statements like 'For contemporary ecosystem processes, it is generally assumed that geologic materials are not an important source of N (Schlesinger, 2013)' in the introduction rather than the discussion can really help people understand why this work is important and novel. As detailed above, we now introduce our ideas about why it is generally assumed that geologic materials are not an important source of N in the Abstract and Introduction.

I found the comparison against different soils rather interesting. The pollution section was less well-developed and without more context on its importance this section might best be removed. Thank you for the kind comment on the soil comparisons. The pollution section is central in supporting our findings for the third research question and is a topic of great relevance in our study region. We have (hopefully!) improved the presentation of the pollution story in response to suggestions from reviewer #1.

**Citation**: https://doi.org/10.5194/egusphere-2024-1165-RC2